# The selenoprotein P 3' untranslated region is an RNA binding protein platform that fine tunes selenocysteine incorporation

**Sumangala P. Shetty, Nora T. Kiledjian, Paul R. Copeland**  *

Department of Biochemistry and Molecular Biology, Rutgers Robert Wood Johnson Medical School, Piscataway, NJ, United States of America

* paul.copeland@rutgers.edu

## Abstract

Selenoproteins contain the 21st amino acid, selenocysteine (Sec), which is incorporated at select UGA codons when a specialized hairpin sequence, the Sec insertion sequence (SECIS) element, is present in the 3' UTR. Aside from the SECIS, selenoprotein mRNA 3' UTRs are not conserved between different selenoproteins within a species. In contrast, the 3'-UTR of a given selenoprotein is often conserved across species, which supports the hypothesis that cis-acting elements in the 3'-UTR other than the SECIS exert post-transcriptional control on selenoprotein expression. In order to determine the function of one such SECIS context, we chose to focus on the plasma selenoprotein, SELENOP, which is required to maintain selenium homeostasis as a selenium transport protein that contains 10 Sec residues. It is unique in that its mRNA contains two SECIS elements in the context of a highly conserved 843-nucleotide 3' UTR. Here we have used RNA affinity chromatography and identified PTBP1 as the major RNA binding protein that specifically interacts with the sequence between the two SECIS elements. We then used CRISPR/Cas9 genome editing to delete two regions surrounding the first SECIS element. We found that these sequences are involved in regulating *SELENOP* mRNA and protein levels, which are inversely altered as a function of selenium concentrations.

## Introduction

The role of untranslated regions (UTRs) in post transcriptional gene regulation is well established. A pointed example of this is the requirement for a specific 3' UTR sequence that directs selenocysteine (Sec) incorporation. Sec is incorporated into specific sites within 25 human proteins at specific UGA codons, which would otherwise signal translation termination. Sec incorporation requires a stem-loop structure in the 3' UTRs of mammalian selenoprotein mRNAs known as a Sec insertion sequence (SECIS) element. While the SECIS element is known to be necessary and sufficient for Sec incorporation, albeit with varying efficiency [1], the role of surrounding sequences in the extremely diverse array of selenoprotein 3' UTRs has only recently been the subject of investigation. For example, the sequence upstream of the selenoprotein S (SELENOS) SECIS element regulates Sec incorporation via two adjacent but non-

**Funding:** NIH NIGMS R01GM077073. The funders had no role in study design, data collection and analysis, decision to publish, or preparation of the manuscript.

**Competing interests:** The authors have declared that no competing interests exist.

**Abbreviations:** PTBP1, polypyrimidine tract binding protein 1; SECIS, selenocysteine insertion sequence; SELENOP, selenoprotein P; SBP2, SECIS binding protein 2; UTR, untranslated region.

overlapping sequences that both positively and negatively impact Sec incorporation efficiency [2]. In terms of cellular context, SELENOS mRNA levels have been reported to change in response to cytokines [3], but a connection to modulation of Sec incorporation by regulatory sequences in the 3' UTR has not yet been made. The potential for other examples of regulation is substantial considering that the size of selenoprotein mRNA 3' UTRs range from ~300 to ~6000 nt [4].

All but one of the human selenoproteins contain one or two Sec codons in their coding region, and a single SECIS element in their 3' UTRs. The exception to this is selenoprotein P (*SELENOP*), which in humans contains 10 Sec codons and 2 SECIS elements in the 3' UTR. The two SECIS elements sit in the context of 843 nt of well-conserved sequence. As such, *SELENOP* provides a unique example to study the role of non-SECIS 3' UTR sequences. As a hepatokine regulator of whole body selenium, *SELENOP* mRNA and protein levels are regulated by a multitude of physiological conditions including insulin levels, exercise and, of course, selenium availability [reviewed in 5]. Since SELENOP supports general selenoprotein synthesis by delivering selenium to the extrahepatic tissues, it is a key player in maintaining cellular selenium homeostasis and redox balance through the myriad of selenoenzymes that resolve oxidative stress. In addition, SELENOP itself has a thioredoxin motif that may directly be involved in maintaining redox balance [6]. Several previous studies have analyzed the effects of deletions in the *SELENOP* 3' UTR but only in the context of transfected cells or in vitro translation, and no function could be ascribed to the non-SECIS portions of the UTR [7–9]. To date, no studies have revealed a specific role of 3' UTR sequences in mediating regulation.

Using RNA affinity chromatography, we have identified the polypyrimidine tract-binding protein (PTBP1) as one of several RNA binding proteins that interacts specifically with conserved regions of the SELENOP 3' UTR. Coupling genomic mutagenesis, UV crosslinking and metabolic labeling with $^{75}$Se, we found that these regions are required for moderating SELENOP expression in response to varying selenium concentration and oxidative stress.

## Material and methods

### CRISPR/Cas9 genome editing

Two guide RNAs against the interSECIS region of SELENOP 3' UTR (Table 1) were synthesized and cloned separately into pSpCas9(BB)-2A-Puro (PX459) V2.0 [10]. This vector was a gift from Feng Zhang (Addgene plasmid # 62988; http://n2t.net/addgene:62988; RRID: Addgene_62988). HepG2 cells were grown to 75% confluence in 10%FBS. 2.5 μg each of plasmid DNAs harboring the SELENOP targeting sgRNAs were electroporated using the Amaxa 4D Nucleofector protocol for HepG2 cells. The pmaxGFP plasmid (0.15 μg) was co-transfected to allow calculation of electroporation efficiency. After 48 hours, GFP positive (top 3%) cells were sorted as single cells into a 96 well plate containing 50% HepG2 conditioned media/ 50% DMEM F12 with 10% fetal bovine serum. The single cell clones were grown for 3 weeks and then split and screened for deletions by $^{75}$Se-selenite labeling. The genome editing was confirmed using genomic DNA PCR with Herculase II enzyme.

### $^{75}$Se labeling and oxidative stress

Cells were grown to 80% confluence and then switched to serum free media supplemented with 100 nM $^{75}$Se-selenite (Research Reactor Center, University of Missouri, Columbia). After 12 hours, the media were collected and centrifuged at 2500 x g for five minutes at 4˚C. The top 80% of the centrifuged media were transferred into a new tube. 1.5% of the total centrifuged media were used for analysis by 12% SDS-PAGE. The adhered cells were then gently washed with cold PBS and lysed in 1% NP-40 lysis buffer (50mM Tris-HCl pH 8.0, 150 mM sodium

**Table 1. Oligonucleotide sequences used in this study.**

| | |
|---|---|
| Δpre sgRNA 5' | ATTGTGTCTAGACTAAATTG |
| Δpre sgRNA 3' | TTTATGATGGAGCAACTGAA |
| Δinter sgRNA 5' | GAGATAAGTAAAGAAAAAAA |
| Δinter sgRNA 3' | GCTGTCTTAAAAGATATAAG |
| SELENOP qPCR Fwd | CTAGGAGCTGATGCTGCCATT |
| SELENOP qPCR Rev | GGTGATTGCAGACCCTGTTTTT |
| Actin qPCR Fwd | GCGCGGCTACAGCTTCA |
| Actin qPCR Rev | CTTAATGTCACGCACGATTTCC |
| Genomic PCR Primer Fwd | ATATTTAAAATAGGACATACTCCCC |
| Genomic PCR Primer Rev | CAGCTTTAAGGTTTTTATTGAATTTATTTG |
| MS2 sequence | CTCGAGACTAGTAGATCTTTTTTTTACTAGTAGATCTTTTTTTGATGAG<br>GATTACCCATCACTAGTAGATCTTTTTTTGATGAGGATTACCCATCACT<br>AGTAGATCTTTTTTTACTAGTAGATCTTTTTTTGATGAGGATTACCCAT<br>CACTAGTAGATCTTTTTTTACTAGTAGATCTTTTTTTCTCGAG |
| RNA-ÂIP Fwd | CAACTGAAAGGTGATTGCAGCTTTTGGT |
| RNA-ÂIP Rev | AGGAGGTCAGGTTTATAGGGTTTGGTT TACC |

chloride, 1% NP-40, Roche Complete protease inhibitor). The lysate was then cleared by centrifugation at 17,000 x g for 10 min at 4°C. For oxidative stress, [75]Se-selenite was added 24 hours prior to lysis and the cells were treated with the indicated range hydrogen peroxide starting six hours prior to lysis. The conditioned media and cell lysates were analyzed as described above.

## Quantitative RT-PCR

Total RNA was purified using the RNAeasy kit (Qiagen). 200 ng of total RNA were converted to oligo dT primed cDNA using SuperScript III (Applied Biosystems). Gene specific forward and reverse primers were designed to amplify 57–71 base long reference and target amplicons (Table 1). Actin was used as the reference target. qRT-PCR was performed using an ABI Step One Plus Real Time PCR System (Applied Biosystems) and PCR Master Mix (ABI) with Power SYBR Green (Invitrogen) and ROX reference Dye (Invitrogen). The total reaction volume was 20 μl with 5 μl of 1:5 diluted RT reaction. Working concentration of the primers in the reaction was 0.25 μM. Thermal cycling conditions were 95°C for 10 min followed by 40 cycles of 95°C for 15 sec, 60°C for 1 min. Melt curve analysis was performed for each sample to ensure a single amplification product. Samples were analyzed in triplicate for both the reference gene and the target gene. Quantitation was performed using the comparative ΔΔCt method. We used actin as the normalizer and the calibrator sample was the endogenous human SELENOP from untreated samples. Primers in the acceptable efficiency range (90–110%) were determined using the standard curve method.

## Immunoblot analysis

For SELENOP immunoblots, 15 μl of the lysate (10% of the total cell lysate which is ~30 μg total protein) as described above were resolved by 12% SDS-PAGE, blotted to nitrocellulose membrane (Amersham Biosciences), blocked in 5% nonfat dried milk for 1 hour at room temperature and probed using a monoclonal horseradish peroxidase-conjugated (HRP) anti-SELENOP antibody (Thermo-Fisher 37A1) at a 1:1,000 dilution overnight. Signal was detected using the SuperSignal West Femto kit (Pierce) according to the manufacturer's protocol. For

PTBP1 immunoblot a mouse monoclonal antibody (Thermo Fisher 32–4800) was used at a 1:1000 dilution.

## Affinity chromatography

The entire rat SELENOP 3' UTR was cloned into the pcDNA3.1 vector using the TOPO-TA cloning kit (Invitrogen). 3 copies of the viral RNA MS2 sequence separated by random sequence were synthesized commercially from IDT technologies (Table 1). These synthetic constructs were ligated downstream of the rat SELENOP 3' UTR sequence at a *Xho I* restriction site. As a control, we also ligated viral MS2 sequence to the 3' end of a non-specific sequence corresponding to a fragment of the coding region for SECISBP2L, which has a similar GC content and is of similar length as SELENOP 3' UTR (39% GC versus 31% for the SELENOP 3' UTR). The SELENOP 3' UTR and the control plasmid were linearized using *Bsb 1* and in vitro transcribed using the Ribomax kit (Promega). The RNA was then purified using p30 size exclusion columns (Bio-Rad) and quantified. The GST-MS2 protein was derived from a clone obtained from Rachel Green (HHMI) and was purified as described earlier [11].

For Bead preparation, 10 µl of magnetic glutathione beads (Pierce) were incubated with 100 µg of purified GST-MS2 protein for 2 hours at 4˚C in buffer A (20 mM Tris-OAc, pH 7.5, 100 mM KOAc, 2 mM DTT, 2.5 mM Mg(OAc)$_2$, 0.25 mM spermidine, 0.4 mM GTP). The protein-bound beads were then washed with buffer A three times (10 minutes each) and then incubated with 10 µg of in vitro transcribed MS2 tagged RNA in buffer A containing 1 U/µl RNAsin for an hour and again washed with buffer A. The beads were then incubated with pre-cleared cell lysate for 2 hours at 4˚C and then washed buffer A three times (10 minutes each) and eluted with Buffer A plus 1 M NaCl for 20 minutes. The elution was sent for LC MS/MS analysis for peptide identification at the Rutgers Biological Mass Spectrometry Facility where MudPIT analysis was performed as previously described [12].

## UV crosslinking

Plasmids containing the SELENOP 3' UTR fragments or mutants were linearized with *Not I* and transcribed with T7 RNA polymerase in the presence of [32P]-α-UTP (Perkin Elmer). Recombinant GST-PTBP1 was incubated with 20 fmol [32P]-α-UTP labeled fragments. Following incubation, complexes were UV irradiated at 254 nm for 10 min on ice and subsequently treated with 20 µg RNase A for 15 min at 37˚C. Samples were resolved by 10% SDS-PAGE, and visualized by phosphorimaging. For the mutants in this study, we replaced the U-rich stretches in the interSECIS with either of the other 3 nucleotides. In addition, we also created a version where most of the U residues in the interSECIS were changed to A (see S1 Fig for sequences).

## Recombinant GST-PTBP1 preparation

GST-PTBP1 was expressed and purified from a construct provided by Lori Covey (Rutgers University). The protein was produced in *E. coli* similar to the GST-MS2 procedure [11] and then purified on a glutathione-Sepharose column (GE Healthcare). Following elution, the purified GST-PTB was dialyzed against dilution buffer (50 mM HEPES, pH 7.6, 1 mM DTT, 1 mM MgCl2 and 20% glycerol) and stored at -80˚C.

## RNA-immunoprecipitation (RNA-IP)

Wild type and CRISPR mutated HepG2 cells were grown in EMEM complete media and collected at 80% confluence. Cells were placed on ice and washed three times with ice cold PBS,

then lysed for 10 min on ice with IP lysis buffer (10 mM Tris-HCl pH 7.6, 150 mM NaCl, 0.5% NP-40, 5% Glycerol and Roche protease inhibitor). Lysates were centrifuged at 17,000 × g for 10 min and supernatant incubated overnight at 4˚C with 2 μg of anti-PTBP1 antibody (monoclonal-ThermoFisher) or anti-FLAG antibody (monoclonal-Sigma). Bound proteins were pulled down using the Dynabeads Protein G Kit (Life Technologies). After pull-down, 10% of beads were boiled for 5 minutes at 95˚C in SDS sample buffer and proteins were resolved on a 12% SDS-PAGE gel. The gel was blotted onto nitrocellulose and probed with 1:1000 anti-PTBP1 antibody. The remaining beads were used for total RNA extraction using Trizol reagent. cDNA synthesis was performed with both gene specific primers and oligo dT and PCR amplified using the RNA-IP primers shown in Table 1.

## Statistical analysis

Gel quantitation was performed using Imagequant IQTL 8.1 using the rolling ball background correction. All experiments were performed as at least 3 biological replicates and significance was determined using the Student's *t*-test. Graphical data shows the means with error bars representing standard deviation.

## Results

### Identification of interSECIS RNA binding proteins

The 843 nucleotide (nt) human *SELENOP* 3' UTR is highly conserved among mammals (76% identity) and as shown in Fig 1 can be divided into 5 regions: 1) the sequence upstream of SECIS-1, hereafter the "preSECIS" region (254 nt); 2) SECIS-1 (83 nt); 3) the sequence between the two SECIS elements, hereafter the "interSECIS" region (358 nt); 4) SECIS-2 (81 nt); 5) the sequence downstream of SECIS-2 (63 nt). Since large, conserved AU-rich 3' UTRs are a hallmark of post-transcriptional regulation, it is very likely that the AU-rich *SELENOP* 3' UTR (overall 71% AU) is targeted by one or more RNA binding proteins. With an initial focus on the sequence between the two SECIS elements, we performed RNA affinity chromatography based on the MS2 RNA/MS2 coat protein interaction [13]. Three MS2-tagged RNAs were used as bait (Fig 2A): 1) full-length rat *SELENOP* 3' UTR, 2) a mutant *SELENOP* 3' UTR with the interSECIS region deleted (ΔinterSECIS) and an unrelated control RNA that had a similar

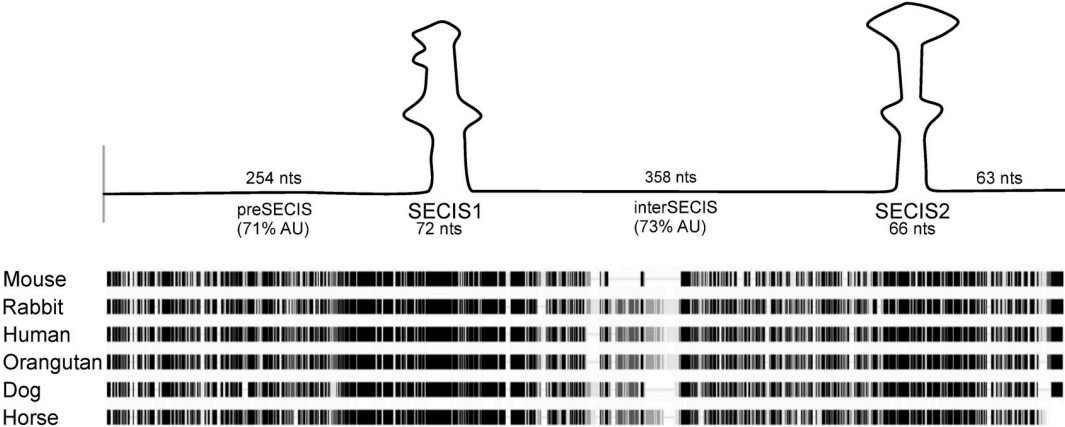

**Fig 1. Conservation in the SELENOP 3' UTR.** A) Diagram and multiple sequence alignment of the mammalian SELENOP 3' UTR. The alignment of sequences from the species indicated was performed using the MUSCLE algorithm, identical positions are black.

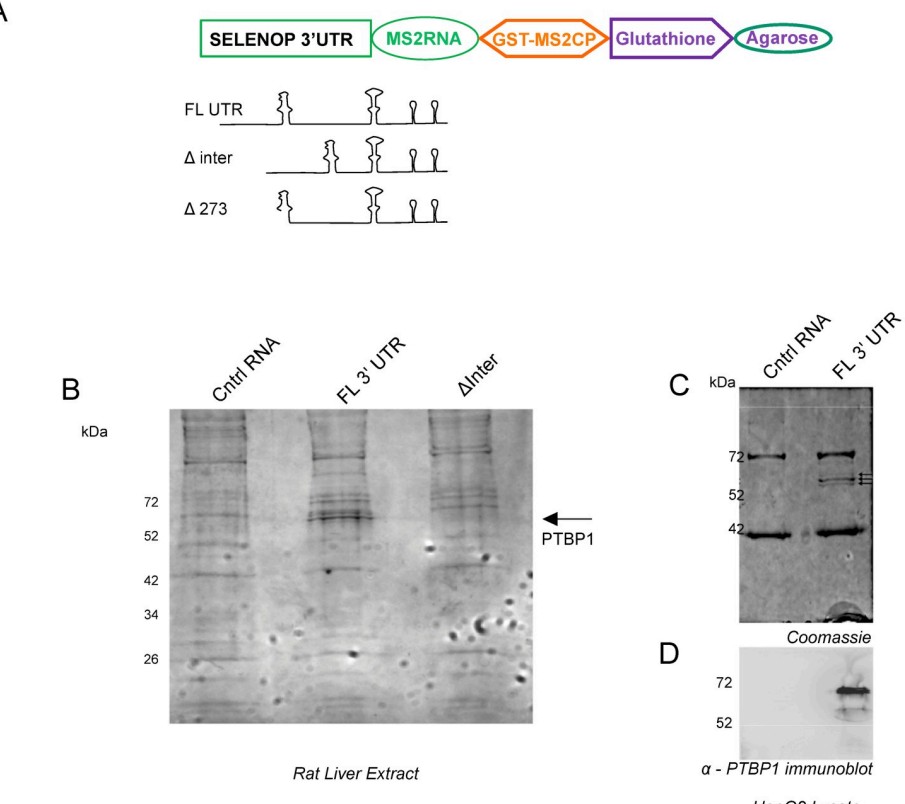

**Fig 2. RNA affinity chromatography identifies PTBP1 and other SELENOP 3' UTR binding proteins.** A) Diagram of the components used for glutathione agarose affinity chromatography and RNA constructs used. B) SDS PAGE analysis of rat liver proteins eluted from a GST-MS2 column bound with either control RNA, SELENOP 3' UTR (FL 3' UTR), and the interSECIS deletion mutant (ΔInter). The gel was stained with Coomassie and the bands of interest excised from the gel for LC MS/MS analysis. C) SDS PAGE and D) immunoblot analysis of proteins eluted from RNA affinity experiments as described in A) except that HepG2 lysate was used. The immunoblot was probed with anti-PTBP1 antibody. E) Peptide counts and cognate genes that were identified by LC MS/MS MudPIT analysis after RNA chromatography using control RNA (Cntl), wild-type SELENOP 3' UTR (WT) and an RNA lacking the first 273 nt of the SELENOP 3' UTR (Δ273). The ratio of peptide counts from WT UTR versus control RNA is shown (WT:Cntr). The genes shown here correspond to those with a ratio of 5 or above and a peptide count of 10 or above.

GC content and the same length. These RNAs were synthesized in vitro and attached to gluta-thione agarose magnetic beads that had been pre-bound with GST-tagged MS2 coat protein, which binds with high affinity to the MS2 RNA sequence tag. The RNA-bound beads were incubated with rat liver extract and the bound proteins were eluted with high salt after exten-sive washing. Fig 2B shows SDS PAGE analysis of the proteins that eluted from the *SELENOP* 3' UTR and control RNA. We observed a series of bands at ~65 kDa that specifically eluted from the wild-type *SELENOP* 3' UTR but not the control RNA or the ΔinterSECIS mutant. To identify these proteins, the bands were excised from the gel and subjected to LC MS/MS. The predominant peptides corresponded to the polypyrimidine tract binding protein 1 (PTBP1). It is likely that the other bands correspond to some of the known PTBP1 isoforms (there are 7 transcript variants listed in the NCBI RefSeq database). To verify the identity of these bands, we repeated the RNA affinity chromatography using human hepatoma cell (HepG2) lysate. Fig 2C shows that we obtained a similar result with three bands in the ~65 kDa range eluting spe-cifically from the *SELENOP* 3' UTR. Immunoblot analysis using anti-PTBP1 monoclonal anti-body confirmed that the eluted proteins are recognized by the antibody (Fig 2D).

In an effort to identify additional *SELENOP* 3' UTR binding proteins, we repeated the RNA affinity chromatography using the control RNA, the full length 3' UTR and also mutant UTR lacking the first 273 nt of the 3' UTR, thus deleting the preSECIS region plus the 5' portion of SECIS-1 (Δ273). These RNAs were incubated with rat hepatoma (McArdle 7777) cell lysate and the entire chromatographic eluate was submitted for LC MS/MS Multidimensional Pro-tein Identification Technology (MudPIT) analysis. Fig 2E shows the list of proteins that showed significant selectivity for the *SELENOP* 3' UTR (those with 10 more peptides detected). Besides the expected recovery of both SECIS binding proteins as well as PTBP isoforms, abun-dant peptides were recovered from other RNA binding proteins. Both RAVER1 and MATR3 are known PTBP1 cofactors, but ELAV1 and RBM47 may be independently interacting with the *SELENOP* 3' UTR. All of the PTBP isoforms and known PTBP interacting proteins [14]; RAVER1 and MATR3) were also recovered by the Δ273 mutant RNA confirming that the interSECIS region is primarily responsible for assembly of the PTBP-dependent RNP. Interest-ingly, there were some proteins identified that appear to be specifically interacting with the preSECIS region (e.g. ELAV1, RBM47, HNRNPLL, CCAR2 and HNRNPK), which are candi-dates for factors that regulate the efficiency of *SELENOP* translation. Overall these results con-firm the initial identification of PTBP1 as a *SELENOP* 3' UTR binding protein, and expand the list of candidates that may play roles in fine tuning SELENOP expression either through mod-ulating translation or mRNA stability. Notably, most of the proteins identified are primarily nuclear, raising the possibility that the *SELENOP* mRNP is assembled in the nucleus prior to export.

### Direct binding of PTBP1 to the *SELENOP* 3' UTR

In order to determine if PTBP1 directly binds to the *SELENOP* 3' UTR, we performed UV crosslinking analysis using purified recombinant GST-tagged PTBP (GST-PTBP1). For this assay we generated [$^{32}$P]-UTP-labeled RNA fragments corresponding to full length *SELENOP* 3' UTR and 4 other versions with targeted substitutions. Fig 3A shows the canonical PTBP1 binding sites (UCUU) in the human *SELENOP* 3' UTR (green arrows), indicating a broad spectrum of potential interaction sites. In this case we are using UV crosslinking as a non-quantitative approach that reveals the location of binding rather than affinity. We sought to narrow down the location of PTBP1 binding by generating three mutated versions of the *SELENOP* 3' UTR with the patches of U-rich sequences changed to random U-free sequences. In addition, we generated a fourth mutant in which nearly all of the U residues in the

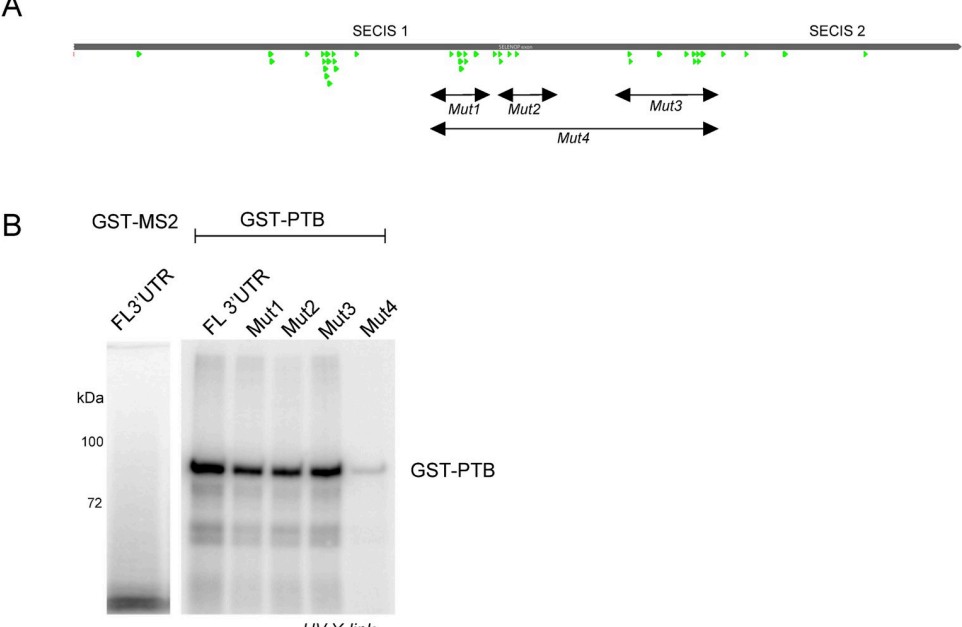

**Fig 3. Recombinant PTBP1 binds directly to the SELENOP 3' UTR.** A) Line diagram of the SELENOP 3' UTR with canonical U-rich PTBP1 binding sites (UCUU) annotated as green arrows. The four regions that were mutated are indicated with arrows. B) SDS PAGE of recombinant GST-PTBP1 protein UV-crosslinked to $^{32}$P-UTP labeled RNA fragments as indicated in A). As a negative control, GST-MS2 coat protein was used as indicated.

interSECIS region were changed to A (Mut1-4, S1 Fig). These fragments were labeled with $^{32}$P UTP and subjected to UV crosslinking analysis with GST-PTB. Fig 3B shows that mutating individual patches of PTB binding sites did not eliminate binding but the alteration of all three sites (Mut 4) significantly reduced crosslinking signal. The low level of signal for Mut4 indicates that the remaining labeled U residues in the preSECIS region are not the primary binding sites for PTBP1. However, the residual signal that is observed in the mut4 lane may represent low affinity interactions in this region. This is consistent with the slight reduction of PTBP1 peptides recovered from the Δ273 mutant in the mass spectrometric analysis shown in Fig 2E. This result establishes that the primary binding site of PTBP1 in vitro is in the interSECIS region rather than at the sites upstream of SECIS 1.

## Deletion analysis using CRISPR/Cas9

In order to reveal regulatory functions for the non-SECIS regions of the *SELENOP* 3' UTR in vivo, we generated genomic deletions of these sequences in HepG2 cells using CRISPR/Cas9. For the preSECIS region we used single guide RNAs (sgRNAs) targeting the sequence 26 nt downstream of the stop codon and 55 nt upstream of SECIS-1. This created a deletion slightly shorter than the Δ273 mutant above due to sgRNA constraints. For the interSECIS region we used a set of sgRNAs targeting the sequence 37 nts downstream of SECIS-1 and 45 nts upstream of SECIS-2 (Fig 4A). The sgRNAs were electroporated along with a GFP vector into HepG2 cells and after 48 hours, the top 3% of fluorescent cells were sorted into a 96-well plate in a single-cell format. Upon confluence, the clonal populations were analyzed for genomic deletion by PCR of genomic DNA. As shown in Fig 4A, sequencing of PCR fragments confirmed that the expected fragments were eliminated, and genomic PCR verified the deletion (Fig 4B, left panel). We also confirmed the genomic deletions by RT-PCR of RNA derived

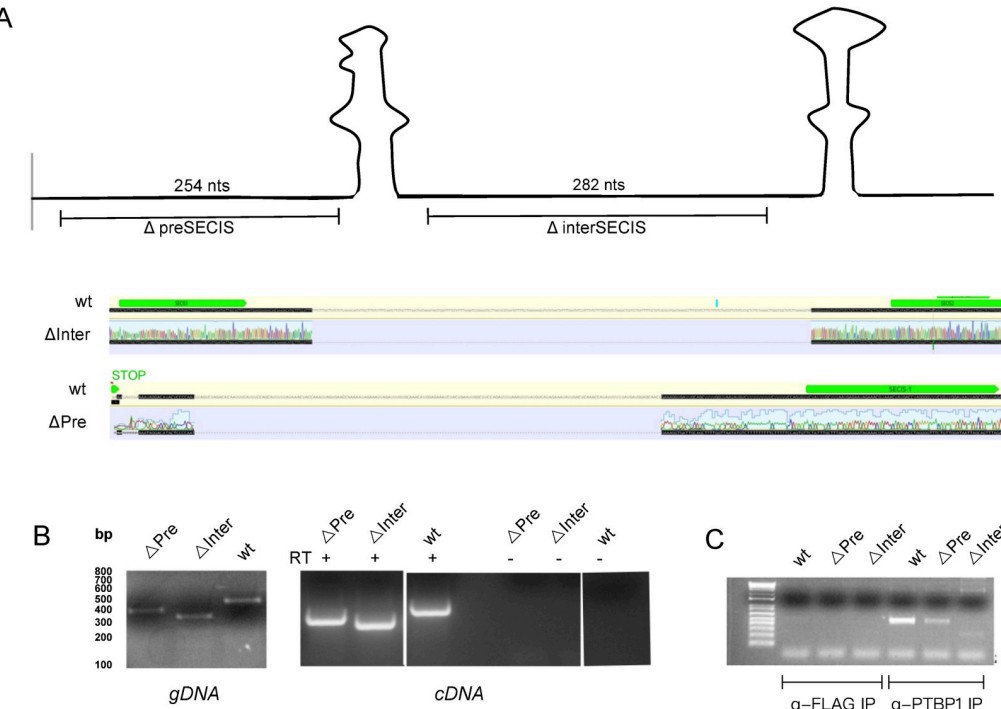

**Fig 4. Editing of the SELENOP gene in HepG2 cells.** A) Diagram of the SELENOP 3' UTR indicating the sequences deleted by CRISPR/Cas9 genome editing. Sequence traces of the deleted regions are shown below. B) genomic PCR (left panel) and RT-PCR (right panel) with primers flanking the deletion sites. C) Lysates from wild-type (WT) and interSECIS deletion (ΔInter) and preSECIS deletion (ΔPre) HepG2 cells were subjected to RNA-IP using the anti-PTBP1 or anti-FLAG control antibodies. RNA was extracted from immunoprecipitates and analyzed for SELENOP mRNA levels by limited-cycle RT-PCR.

from the deletion clones (Fig 4B, right panel). To determine whether the mutated versions of *SELENOP* mRNA were compromised in their ability to bind PTBP1, we performed RNA immunoprecipitation (RNA-IP). HepG2 lysates from wild-type and mutated lines were incubated with anti-PTBP1 antibody or anti-FLAG antibody as a control. RNA was extracted from the immunoprecipitated material and limited-cycle RT-PCR was performed to estimate the recovery of *SELENOP* mRNA. Fig 4C shows that *SELENOP* mRNA was easily detectable from the wild-type and to a lesser extent from the ΔpreSECIS sample, but it was barely detectable from the ΔinterSECIS sample. While this assay is only semi-quantitative, it illustrates a substantial defect in PTBP1 binding in the ΔinterSECIS cells.

## Role of the non-SECIS regions during oxidative stress

Considering the role of selenoproteins in responding to oxidative stress, we hypothesized that SELENOP expression would be modulated by peroxide stress as has been shown for other selenoproteins [15]. We chose peroxide stress as a tool for inducing ROS in mammalian cell culture since it is a well-established method to induce stress in HepG2 cells [16]. To assess the effect of peroxide stress on SELENOP expression, we treated wild type and deletion mutant HepG2 cells with a range of hydrogen peroxide concentrations (5–1000 μM) in the presence of [75]Se-selenite. Fig 5A shows immunoblot and phosphorimage analysis of the conditioned media. Note, in most cell types, two forms of SELENOP are expressed from the same mRNA: the full-length protein and a truncated version resulting from translation termination at the 2nd UGA (Sec) codon. These migrate at 62 and 55 kDa, respectively, due to glycosylation. In

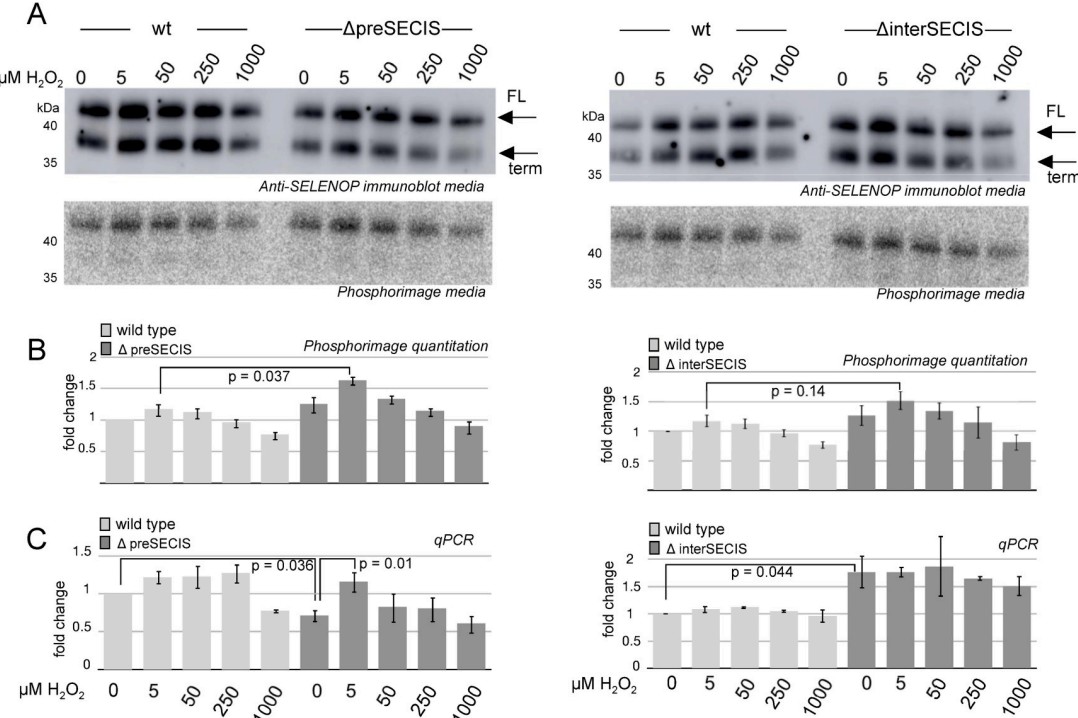

**Fig 5. Hydrogen peroxide treatment reveals a regulatory role for the interSECIS sequence.** A) Wild-type (wt) or interSECIS deletion mutant HepG2 cells were treated with the hydrogen peroxide concentrations indicated and 100 nM [75]Se-selenite for 6 hours. 30 μl of conditioned medium was analyzed by SDS PAGE followed by immunoblot (top) or phosphorimaging (bottom). Full length (FL) and truncated (term) SELENOP resulting from early termination at the second UGA codon are indicated with arrows. B) Quantitation of the band corresponding to full length (FL) SELENOP from the phosphorimage normalized to quantitation of bands from the stained gel. C) Total RNA was isolated from these cells and analyzed by qRT-PCR normalized to actin and the wt set to 1. For B and C, data were plotted as the mean with error bars showing standard deviation. A Student's t-test was used to calculate the p values shown on three biological replicates.

order to improve electrophoretic resolution of the SELENOP bands, the medium was treated with PNGAseF, which removes both N- and O-linked glycosylation. After such treatment, the full length and truncated versions of SELENOP migrate at 42 and 37 kDa respectively (Fig 5A). Quantification of the radiolabeled SELENOP revealed a slight but consistent increase in full length protein expression in both the preSECIS and interSECIS deletion mutants, particularly at the 5 μM peroxide level (Fig 5B). Although this difference was observed for both mutants, only deletion of the preSECIS region resulted in a statistically significant increase in SELENOP expression at 5 μM peroxide treatment. While we did not attempt to perform quantitative immunoblot analysis, the blots revealed that the ratio of full length to truncated SELENOP did not significantly decrease as a result of peroxide treatment, and the mutations did not cause a substantive change in the ratio. In addition, the peroxide treatment did not affect intracellular selenoprotein synthesis as shown by examining the [75]Se-selenite-labeled cell lysates (S2 Fig). These results indicate that the pre- and interSECIS sequences may be required to fine-tune SELENOP expression but that SELENOP expression is not substantively impacted by peroxide exposure under these conditions.

In order to determine the contribution of RNA concentration to the change in SELENOP expression, we performed qRT-PCR. Fig 5C shows that the deletions had opposing effects where the preSECIS deletion caused a ~25% reduction in mRNA levels while the interSECIS deletion caused ~75% increase. These opposing effects indicate that the efficiency of SELENOP

protein production is increased when the preSECIS sequence is deleted and decreased for the interSECIS deletion. Interestingly, we reproducibly observed a spike of ΔpreSECIS mRNA expression at the 5 μM dose of peroxide but this spike was not observed for the ΔinterSECIS mutant mRNA. This finding further supports a complex interplay between regulation of mRNA levels and translation efficiency to allow SELENOP regulation.

## Selenium supplementation reveals a role for PTB binding sites in regulating translational efficiency

In addition to peroxide stress, we also analyzed the effect of altered selenium concentrations. Since standard HepG2 culture conditions are typically selenium deficient, we first determined the effect of cold selenium supplementation on SELENOP expression. Fig 6A shows

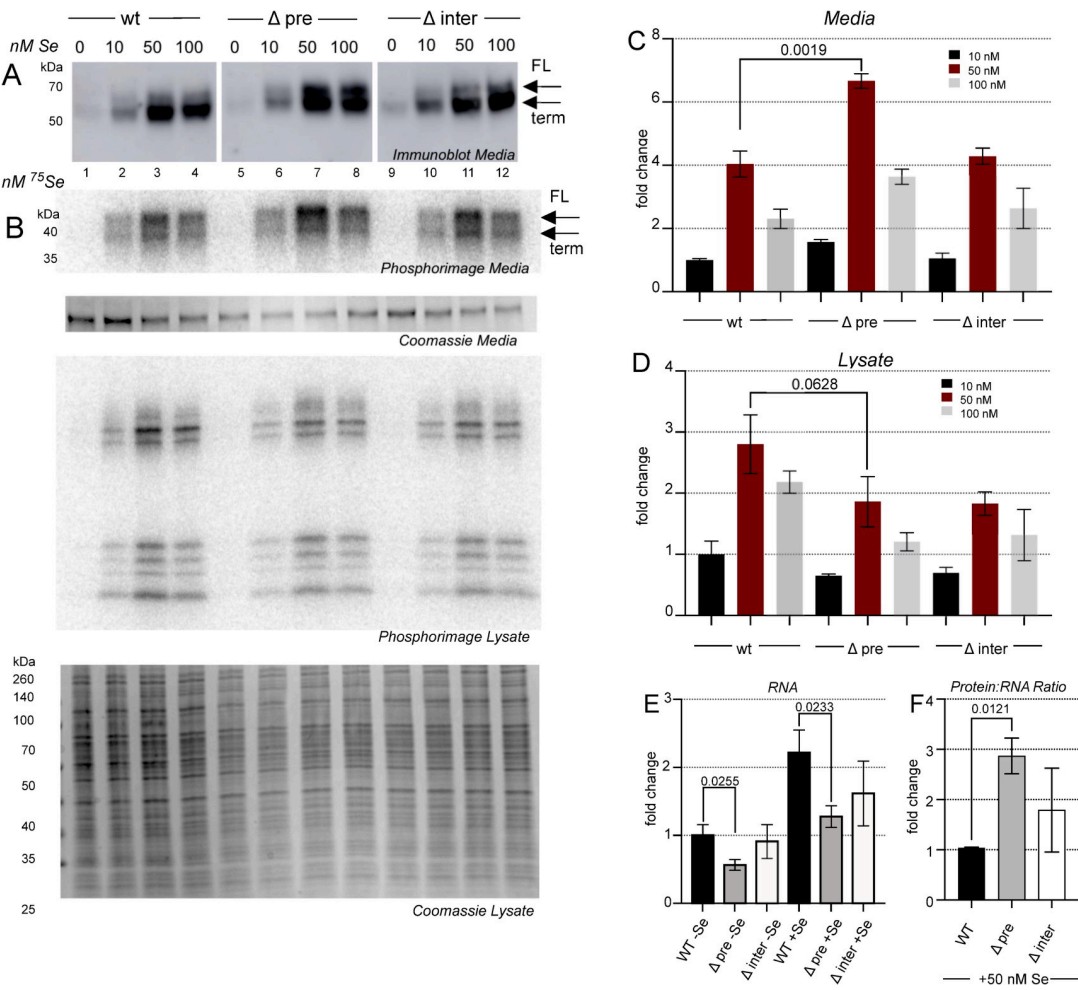

**Fig 6. Limiting selenium reveals a role for the PreSECIS region.** A) HepG2 cells were incubated with increasing concentrations of sodium selenite as indicated. 30 μl of conditioned medium was analyzed by SDS PAGE followed by immunoblot probed with anti-SELENOP antibody. B) Same as in A) except [75]Se-selenite was used to supplement. SDS-PAGE of conditioned media (top panel) or lysate (bottom panels) was analyzed by phosphorimage analysis and Coomassie Stain (top panel), normalized to quantitation of bands from the stained gel. C) Quantitation of the phosphorimage data derived from conditioned media. D) Quantitation of all radioactive bands in the phosphorimage data derived from lysate. E) Total RNA was isolated from the cell lines indicated with and without 50 nM selenium supplementation and analyzed by qRT-PCR normalized to actin and the wt set to 1. F) A plot of the inferred translational efficiency based on the RNA:protein ratio normalized to wild-type. For all quantification, a Student's t-test was used to calculate the p values shown on three biological replicates.

immunoblot analysis of SELENOP over a range of selenium supplementation from 10 to 100 nM. We observed a striking increase in SELENOP expression that was similar in the wild-type and mutated cell lines, highlighting the fact that standard culture conditions are extremely selenium deficient. Additionally, we observed a significant increase in full length SELENOP production from the ΔpreSECIS versus wild-type cell line (compare lanes 3 and 7). To get a more quantitative picture of the effect of selenium supplementation on the amounts of full length and truncated SELENOP, we used $^{75}$Se-selenite in varying amounts. Interestingly we observed a peak of labeling at 50 nM (Fig 6B), likely due to this being the optimal specific activity for detecting incorporation of labeled selenium while the overall concentration is still limiting (i.e., all or most of the supplemental selenium is being incorporated at the 50 nM concentration). When comparing the amount of labeling between wild-type and mutant versions of SELENOP, we observed about 50% more total signal in the ΔpreSECIS mutant than in the wild-type or ΔinterSECIS mutant cells at the 50 nM level (Fig 6C). Analysis of the lysate did not reveal any statistically significant changes in incorporation that correlated with either of the mutations (Fig 6B, lower panel and Fig 6D). Interestingly, the corresponding RNA analysis by qRT-PCR revealed a significant drop in RNA levels for the ΔpreSECIS mutant cells (Fig 6E) such that the inferred translational efficiency (protein:RNA ratio) was almost 3 times higher for the ΔpreSECIS condition (Fig 6F). These results further reveal roles for the non-SECIS regions of the *SELENOP* 3' UTR in fine tuning SELENOP expression.

## Discussion

All selenoprotein 3' UTRs contain an approximately 100 nt SECIS element required for recoding a UGA to allow Sec incorporation. However, most selenoprotein 3' UTRs contain long sequences adjacent to the SECIS element with no known function. It is therefore likely that selenoprotein 3' UTRs contain regulatory motifs in their 3' UTRs. Such motifs in a broad spectrum of mammalian mRNAs have been revealed over the last several decades as key regulators of translation, mRNA localization and mRNA decay [reviewed in 17]. Naturally, RNA binding proteins are key players in most 3' UTR-mediated regulatory processes, many serving as a platform for highly complex ribonucleoprotein complexes. One of the many pleiotropic RNA binding proteins that falls in this category is polypyrimidine tract binding protein (PTBP1), which has demonstrated roles in regulating pre-mRNA splicing, translation, NMD resistance and mRNA stability [reviewed in 18]. Here, we have found that the *SELENOP* 3' UTR is a platform that binds PTBP1 and other RNA binding proteins that are likely playing a role in regulating the efficiency of SELENOP synthesis.

The human *SELENOP* mRNA contains multiple U-rich sequences, many of which are canonical PTBP1 binding sites (Fig 3). The fact that these sites cluster in the region upstream and downstream of SECIS-1 is consistent with our experimental determination that PTBP1 directly interacts with these regions, although the sites upstream of SECIS1 do not seem to play a major role in direct PTBP1 binding. The genomic deletion of regions containing PTBP1 binding sites revealed potential roles in regulating translation since we observed significant changes in the protein/mRNA ratios. While we cannot rule out transcriptional and protein stability components to this result, a role in translation regulation is consistent with prior roles assigned to PTBP1 outside of its canonical role in pre-mRNA splicing. The observed change in protein/mRNA ratios was driven to some extent by changes in steady state mRNA levels with preSECIS deletion causing a decrease in mRNA and interSECIS causing a general increase. Interestingly, we observed a disconnect between the demonstrated PTBP1 binding sites in the interSECIS region and the stronger effects on expression when the preSECIS region was deleted. This likely points to the importance of other RNA binding proteins in the 5' portion

of the 3' UTR that may function independently of PTBP1. This point also underlies our decision not to pursue PTBP1 loss of function studies, which would be impossible to interpret given its ubiquitous roles in multiple cellular processes. Although the magnitude of effects observed were relatively small, the reproducible changes observed as a result of peroxide exposure suggest that stress response is an important aspect of SELENOP regulation. We expect future work to reveal conditions where PTBP1 function is required to regulate SELENOP levels in vivo. For example, *SELENOP* mRNA levels have been reported to dramatically increase in response to exercise stress in mice fed a normal diet but not those fed a high fat diet [19]. We expect the molecular pathways that regulate such a response to work at least in part through the mechanisms uncovered in this report. It is also important to consider the potential for the regulation of PTBP1 levels or isoforms in the context of its SELENOP-related role. Although we are unaware of any known direct correlations between SELENOP and PTBP1 regulation, the pleiotropism of PTBP1 function and regulation was recently delineated in a single cell-type (HEK293), revealing profound complexity and potential for coordinate regulation [20].

This study has also revealed a role for the non-SECIS sequences in response to varying selenium levels. As the major carrier of blood-borne selenium, SELENOP synthesis is necessarily tightly linked to the available selenium concentration. Notably, we observed that the ΔpreSECIS mutation caused a ~2-fold reduction in steady state mRNA levels with a concomitant and unexpected 50% increase in protein production. This apparent increased translational efficiency may suggest that the efficiency of SELENOP production is kept at a moderate level in order to balance the consumption of limiting selenium to allow adequate intracellular selenoprotein production. Alternatively, the attenuation of SELENOP expression as a function of the preSECIS region may be required to maximize the processivity of multiple Sec incorporation events, considering the prior work that showed significant ribosomal pausing at the first UGA codon [8, 21]. Since we also observed a loss of stead-state SELENOP mRNA levels when the PTBP1 binding sites were compromised, we must also pursue the possibility that the PTB RNP may play a role in preventing the recruitment of NMD factors, which are known to play a role in regulating some selenoprotein mRNAs when translation efficiency is compromised [22].

It will be challenging to determine the mechanism of action for these sequences because extensive analysis of SELENOP synthesis in vitro and in transfected cells has failed to reveal a role for the sequences surrounding the SECIS elements [9, 23]. Considering that the results obtained here indicate a regulatory role for the region surrounding SECIS-1, it is likely that the functional RNP complex may form co-transcriptionally and coincident with RNA splicing. Considering the high concentration of PTBP1 in the nucleus and its role in splicing, it is logical that it binds to *SELENOP* pre-mRNA and stays associated to be exported in a conformation that is able to respond to cellular conditions. In the case of oxidative stress, it is possible that the complex constituents are altered, allowing regulated SELENOP expression. The idea that an active *SELENOP* mRNP may form during pre-mRNA processing may explain why we have been unsuccessful in trying to express detectable protein from transfected cDNA [24]. One of the most likely mechanisms by which 3' UTR sequences would affect selenoprotein expression is by modulating SECIS function, either by blocking or possibly even enhancing SBP2 access or affinity. The proximity of the PTBP1 binding sites to SECIS-1, which is sufficient to support full length SELENOP synthesis in vitro and in transfected cells [9], certainly supports this hypothesis. However, if SECIS access were a general mechanism of regulation, then one would expect the sequences and binding proteins to be conserved. The stark lack of conservation among selenoprotein 3' UTR sequences strongly suggests that independent mechanisms evolved to respond to very specific regulatory demands.

## Supporting information

**S1 Fig. Sequence alignments to illustrate the mutations made for UV-crosslinking analysis in Fig 3.**
(TIF)

**S2 Fig. *SELENOP* 3' UTR mutations do not affect intracellular selenoprotein production.**
Cell lysates derived from the $^{75}$Se-selenite labeling described for Fig 5 were analyzed by SDS-PAGE followed by phosphorimage analysis.
(TIF)

**S1 Raw images. This file contains raw uncropped images used to generate the figures in this manuscript.** The numbers below gels correspond to loading order shown in figures.
(PDF)

## Acknowledgments

We acknowledge technical support from Joanna Duong. Thanks to Dr. Lori Covey (Rutgers Department of Cell Biology and Neuroscience) for providing the plasmid encoding GST-PTBP1.

## Author Contributions

**Conceptualization:** Sumangala P. Shetty, Paul R. Copeland.

**Data curation:** Sumangala P. Shetty.

**Formal analysis:** Sumangala P. Shetty, Nora T. Kiledjian, Paul R. Copeland.

**Funding acquisition:** Paul R. Copeland.

**Investigation:** Sumangala P. Shetty, Nora T. Kiledjian, Paul R. Copeland.

**Methodology:** Sumangala P. Shetty, Nora T. Kiledjian.

**Supervision:** Sumangala P. Shetty.

**Validation:** Sumangala P. Shetty.

**Visualization:** Sumangala P. Shetty, Paul R. Copeland.

**Writing – original draft:** Paul R. Copeland.

**Writing – review & editing:** Sumangala P. Shetty, Nora T. Kiledjian, Paul R. Copeland.

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
