## [Decision Letter · Decision Letter 0]

7 Jun 2022

PONE-D-22-10327The Selenoprotein P 3' untranslated region is an RNA binding protein platform that fine tunes selenocysteine incorporationPLOS ONE

Dear Dr. Copeland,

Thank you for submitting your manuscript to PLOS ONE. After careful consideration, we feel that it has merit but does not fully meet PLOS ONE’s publication criteria as it currently stands. Therefore, we invite you to submit a revised version of the manuscript that addresses the points raised during the review process. Your manuscript has been reviewed by three experts. All of them found it interesting and it merits publication after responding to their concerns. Please revise your manuscript according to their suggestions.

One major concern about it is, as one of the reviewers pointed out, there might be a potential duplication of images, phospho imager images of Figure S2. Please investigate this issue, correct it, if necessary, and report how this incident occurred in the revised manuscript. 

We look forward to receiving your revised manuscript.

Kind regards,

Hodaka Fujii, M.D., Ph.D.

Academic Editor

PLOS ONE

Journal Requirements:

"This work was supported by the National Institutes of Health [GM077073 to P.R.C]"

"No, The funders had no role in study design, data collection and analysis, decision to publish, or preparation of the manuscript."

Additional Editor Comments:

Please correct the following typos:

p. 7, the 2nd last line of the 1st section: "shwon" to "shown".

Reviewers' comments:

Reviewer's Responses to Questions

**Comments to the Author**

1. Is the manuscript technically sound, and do the data support the conclusions?

Reviewer #1: Yes

Reviewer #2: Yes

Reviewer #3: Yes

2. Has the statistical analysis been performed appropriately and rigorously? 

Reviewer #1: Yes

Reviewer #2: Yes

Reviewer #3: Yes

3. Have the authors made all data underlying the findings in their manuscript fully available?

Reviewer #1: Yes

Reviewer #2: Yes

Reviewer #3: Yes

4. Is the manuscript presented in an intelligible fashion and written in standard English?

Reviewer #1: Yes

Reviewer #2: Yes

Reviewer #3: Yes

5. Review Comments to the Author

Reviewer #1: PONE-D-22-10327

The Selenoprotein P 3' untranslated region is an RNA binding protein platform that fine tunes selenocysteine incorporation

The manuscript is very well written, the experiments are appropriate and effective. Results convincing, statistical analyses appropriate. Nice to see using both rat and human cell models used, and that extra bands were validated.

There are only very minor comments:

Suggest italicizing SELENOP to indicate the gene/nucleotide sequence and non-italicized when referring to the protein, as this would provide clarity. It appears to be used once on p. 6, but inconsistently so. Technically, only human gene would be SELENOP, whereas mouse and rat would be selenop.

Please change uM to µM (e.g., bottom of p. 8 and a couple of instances within the text on p. 9 and in Figure 5), and ug to µg (p. 16), and uL to µL (p17).

Please bracket ‘respectively’ by commas (p. 9), and change ‘media was’ to either ‘medium was’ or ‘media were’ (on p. 9 and also a couple of times in the methods section, and in legends for Figure 5 & 6)

Insert comma after i.e. (p. 10)

Be consistent in use of either UTRs or UTR’s. Both are used (see first few sentences of discussion) interchangeably. Similarly, spacing between 3’UTR and 3’ UTR should be consistent.

End of first paragraph of discussion: Here, we have found…… (comma missing)

P 15: 200 ng … were….

P 16: 15 µL…. were…

P, 18: E. coli should be italicized

p-values should always be with the zero preceding the period (e.g., p=0.037 in figure 5B, not p=.037)

The only actual questions concerns Figure S2 – can the authors please verify that the same images weren’t accidentally used for both (left and right) or that we are not misinterpreting the use of it? The images seem identical, down to the same artifacts in the phosphor imager bands that are slightly broken the same way.

Reviewer #2: This is a very interesting manuscript that deepens our knowledge of the regulatory frame of selenoprotein P (SELENOP) translation, an intriguingly unique mRNA that contains 2 SECIS elements that direct insertion of up to 10 selenocysteines in the polypeptide chain. The authors identified for the first time an RNA binding protein, PTBP1, that binds to an intermediate region between the 2 SECIS elements and regulates translation efficiency of this mRNA. The manuscript is clearly written, and the logical flow of findings is very straightforward to follow, with the adequate experiments needed and appropriate controls. It will add valuable information to the field of selenoprotein translation. Nevertheless, some minimal comments below would, if addressed, enrich further the manuscript quality prior to publication.

COMMENTS

Introduction:

• "All but one of the human selenoproteins contain a single Sec codon in their coding region" - There is a second UGA sometimes recognized as Sec in Deiodinase type 2 (see PMID: 11425850). Please rephrase.

Methods:

• Has the SELENOP antibody used in this manuscript been validated?

• Statistical Analysis used in experiments is embedded in figure legends. It will be best organized if shown as a subsection of the Methods.

Discussion:

• "For example, SELENOP mRNA levels have been reported to dramatically increase in response to exercise stress in mice fed a normal diet but not those fed a high fat diet [16]." Is PTBP1 regulated by exercise? A discussion on common potential regulators of PTBP1 should be incorporated to enrich the discussion.

• "(...) it is likely that the 14 functional RNP complex may form co-transcriptionally and coincident with RNA splicing." Has any of the proteins recognized by the screening related to RNA splicing that are below the threshold chosen for the selection of peptides? This could be an interesting point to discuss further.

• This manuscript showed that mutation in the preSECIS1 sequence reduces translational efficiency. PTBP1 could, via binding to an NMD partner, contribute also to nonsense-mediated decay of SELENOP mRNA according to Se conditions. This possibility would also enrich the discussion.

Figures:

• Legend of Fig. 2 has "D" when it should be "E" for the peptide counts data.

Reviewer #3: This study reports investigation of conserved sequences in the 3’UTR of selenoprotein P, using RNA affinity chromatography and mass spec. The overall premise is sound, and the experimental rationale is well thought out, with appropriate controls.

The authors report identification of PTBP1 as the major RNA binding protein specifically interacting with the inter-SECIS region. They further use CRISPR/Cas9 genome editing to assess functional role, and report regulation in response to oxidative stress and selenium concentration.

They further report that the ΔpreSECIS mutation caused a ~2-fold reduction in steady state mRNA levels with a concomitant increase in protein production. This intriguing finding highlights avenues for future investigation into the complex regulatory mechanisms required for expressing this critically important but translationally challenging and perplexing protein.

A minor point that should be corrected on p. 7, line 12 – mutating is misspelled in this sentence, “Figure 3C shows that mutatng individual patches…”

6. PLOS authors have the option to publish the peer review history of their article (what does this mean?). If published, this will include your full peer review and any attached files.

Reviewer #1: No

Reviewer #2: No

Reviewer #3: No

---

## [Author Response · Author response to Decision Letter 0]

13 Jun 2022

Dear Dr. Fujii,

Thank you for taking the time to handle the review of our manuscript (PONE-D-22-10327

The Selenoprotein P 3' untranslated region is an RNA binding protein platform that fine tunes selenocysteine incorporation). Below please find responses to the points raised by reviewers.

First and foremost, to address the issue of the duplicated image. Many thanks to Reviewer #1 for noticing this. Yes, I can confirm and was horrified that the gel images for the right side of figure S2 (right panel) are a duplicate of the gels in the left side. As the reviewer pointed out, the duplication is readily apparent, and although I don’t know how it escaped my attention during the submission process, I will explain how it likely happened. The workflow for manuscripts from my lab is that the person generating data puts “rough” labeled images as a page in the illustration program we use, which is called Sketch. It is then my job to create the final labeled figures for submission. In this case I went back to the original Sketch file that the first author (Suma Shetty) put together and found that the two panels for Figure S2 started off as two separate figures. What likely happened is that I created the labels for one image, and went back and accidentally grabbed the same images instead of going to the other figure. I would be happy to provide the original Sketch file if that is desired. In any event, I hope it is obvious that there was no intent to deceive as the corrected data reveals the same result.

Regarding the funding statement, the current wording is correct: "No, The funders had no role in study design, data collection and analysis, decision to publish, or preparation of the manuscript."

Funding Information: NIH NIGMS R01GM077073

Regarding Data Availability, the minimal data set underlying the results described in the manuscript are fully contained with the main and supplementary figures.

Reviewer #1:

Thank you for the insightful comments and careful reading. All of the suggested corrections were made (proper gene nomenclature for SELENOP, units and punctuation).

See above for the issue with duplicated images for Figure S2

Reviewer #2:

Again, thank you for the insightful comments and careful reading. All of the suggested minor corrections were made.

Regarding the SELENOP antibody validation, it is a commercial antibody that the manufacturer has stated was validated with purified human selenoprotein P. In addition, there are two validating features of the published data: 1) the bands detected only appear when adequate selenium is included in the medium (Figure 6A), and those bands resolve at their expected molecular weights before (Figure 6A) and after (Figure 5A) PNGAse F treatment.

Statistical methods have been added to the methods section.

A statement about the potential for coordinate PTBP1 regulation has been added to page 13 (we could not find any evidence for directly correlated regulation, but instead point to a very thorough analysis of PTBP1 isoforms and their regulation from Jamie Cate’s group in 2019).

Regarding the potential for having identified additional splicing factors in the mass spec data, unfortunately it is not really possible to extrapolate function from this group as there are hundreds of proteins encompassing many cellular processes that are below the cutoff. 

Regarding the mention of PTBP1 regulating NMD, this is an excellent suggestion that we have added to the Discussion.

Reviewer #3:

Again, thank you for the insightful comments and careful reading. All of the suggested corrections were made.

---

## [Decision Letter · Decision Letter 1]

1 Jul 2022

The Selenoprotein P 3' untranslated region is an RNA binding protein platform that fine tunes selenocysteine incorporation

PONE-D-22-10327R1

Dear Dr. Copeland,

We’re pleased to inform you that your manuscript has been judged scientifically suitable for publication and will be formally accepted for publication once it meets all outstanding technical requirements.

Kind regards,

Hodaka Fujii, M.D., Ph.D.

Academic Editor

PLOS ONE

Additional Editor Comments (optional):

Reviewers' comments:

Reviewer's Responses to Questions

**Comments to the Author**

1. If the authors have adequately addressed your comments raised in a previous round of review and you feel that this manuscript is now acceptable for publication, you may indicate that here to bypass the “Comments to the Author” section, enter your conflict of interest statement in the “Confidential to Editor” section, and submit your "Accept" recommendation.

Reviewer #1: All comments have been addressed

Reviewer #2: All comments have been addressed

Reviewer #3: All comments have been addressed

2. Is the manuscript technically sound, and do the data support the conclusions?

Reviewer #1: Yes

Reviewer #2: Yes

Reviewer #3: Yes

3. Has the statistical analysis been performed appropriately and rigorously? 

Reviewer #1: Yes

Reviewer #2: Yes

Reviewer #3: Yes

4. Have the authors made all data underlying the findings in their manuscript fully available?

Reviewer #1: Yes

Reviewer #2: Yes

Reviewer #3: Yes

5. Is the manuscript presented in an intelligible fashion and written in standard English?

Reviewer #1: Yes

Reviewer #2: Yes

Reviewer #3: Yes

6. Review Comments to the Author

Reviewer #1: Very interesting paper, and the revisions look good. Thank you for taking the time to verify the image source.

Reviewer #2: The Authors have addressed all comments appropriately. Without further issues to be raised, this manuscript seems acceptable to publication into PLOS ONE.

Reviewer #3: (No Response)

7. PLOS authors have the option to publish the peer review history of their article (what does this mean?). If published, this will include your full peer review and any attached files.

Reviewer #1: No

Reviewer #2: No

Reviewer #3: No

---

## [Editor Report · Acceptance letter]

8 Jul 2022

PONE-D-22-10327R1 

The Selenoprotein P 3' untranslated region is an RNA binding protein platform that fine tunes selenocysteine incorporation 

Dear Dr. Copeland:

I'm pleased to inform you that your manuscript has been deemed suitable for publication in PLOS ONE. Congratulations! Your manuscript is now with our production department. 

Kind regards, 

on behalf of

Dr. Hodaka Fujii 

Academic Editor

PLOS ONE